# Boomtown Urbanization and Rural-Urban Transformation in Mining and Conflict Regions in Angola, the DRC and Zambia

Cristina Udelsmann Rodrigues [1,*], Patience Mususa [1], Karen Büscher [2] and Jeroen Cuvelier [2]

1   Nordic Africa Institute, 75147 Uppsala, Sweden; patience.mususa@nai.uu.se
2   Conflict Research Group, Ghent University, 9000 Gent, Belgium; karen.buscher@ugent.be (K.B.); jeroen.cuvelier@ugent.be (J.C.)
*   Correspondence: cristina.udelsmann.rodrigues@nai.uu.se; Tel.: +46-0701679661

**Abstract:** Starting from temporary settlements turning into permanent urban centers, this paper discusses the transformations taking place through the process of so-called 'boomtown' urbanization in Central and Southern Africa. Based on data collected in Angola, Zambia and the Democratic Republic of Congo, the paper identifies the different conditions for migration and settlement and the complex socio-economic, spatial, as well as political transformations produced by the fast growth and expansion of boomtowns. Different historical and contemporary processes shape boomtown urbanization in Africa, from colonial territorial governance to large- and small-scale mining or dynamics of violence and forced displacement. As centers of attraction, opportunities, diversified livelihoods and cultures for aspiring urbanities, boomtowns represent an interesting site from which to investigate rural-urban transformation in a context of resource extraction and conflict/post conflict governance. They equally represent potential catalyzing sites for growth, development and stability, hence deserving not only more academic but also policy attention. Based on the authors' long-term field experience in the countries under study, the analysis draws on ethnographic fieldwork data collected through observations as well as interviews and focus group discussions with key actors involved in the everyday shaping of boomtown urbanism. The findings point to discernible patterns of boomtown consolidation across these adjacent countries, which are a result of combinations of types of migration, migrants' agency and the governance structures, with clear implications for urban policy for both makeshift and consolidating towns.

**Keywords:** temporary settlement; protracted settlement; boomtown urbanization; urban emergence; urban growth; Africa; Angola; DRC; Zambia

## 1. Introduction

This article presents a multiple case study analysis of boomtown urbanization in the three adjacent countries in Sub-Saharan Africa, Angola, the Democratic Republic of Congo and Zambia. The joint analysis of these three case studies of urbanization brings together research on migration and urbanization and highlights the varied dynamics of agglomeration and town building in a context of mining and voluntary or forced displacements and 'emplacements'.

Based on qualitative data collected through interviews, observations and focus-group discussions with different urban stakeholders across these countries, the article examines transformations taking place throughout the process of so-called 'boomtown' urbanization. The countries' shared historical, political, regional and socio-economic particularities makes the comparison of their intertwined mining and urbanization processes relevant. Mineral extraction and migration in Angola, the DRC and Zambia has been a key economic driver for rural-urban transformation, and informs the selection of these cases for analysis, as does their similar postcolonial political trajectories.

By focusing on the impact of the intertwined dynamics of resource exploitation and political instability or violent conflict on the emergence and consolidation of new urban

centers, the article presents insights into: (i) the dynamics leading to town booming; (ii) the consolidation trajectories defining temporariness or permanency of urbanity and (iii) the political characteristics of urbanization in a context of war and violence. It demonstrates how, in these cases, different conditions for migration, settlement and town building were created over the years in their mining regions. The article advances by examination on the complex socio-economic, spatial, as well as political transformations produced by the fast growth and expansion of boomtowns and how they are imbricated. The analysis is relevant both in the context of the region and of migration and urbanization studies: both phenomena are understudied. The unpredictable dynamics of these phenomena, particularly in relation to their setting and the outcomes and effects on population dynamics, urban planning and management or the environment, make their study pertinent. In this regard, beyond academic relevance, the article's approach aims to provide substance to policy and governance debates and implications: First, it brings forward the importance of looking at country case studies as integrated in regional Sub-Saharan Africa's dynamics and, second, of looking at the intertwinement of mining and urbanization. For researchers as well as policy makers, a closer investigation into these fast-growing agglomerations is key, as they are normally not considered attractive poles of investment and livelihood diversification on a local as well as on a regional scale. Moreover, their rapid transformation points to their strong potential to catalyze processes of positive development, social change, growth and stability. As it stands, they garner less attention than the larger primary urban agglomerations—as a result, often not much is known about their settlement patterns and consolidation, nor their social and economic networks or political characteristics.

## 2. Literature Review and Theoretical Framework

The concept of boomtown urbanization refers to a sudden and rapid spatial, demographic and economic growth of rural regions driven by changing economic regimes and people's search for livelihood opportunities [1,2]. Boomtown urbanization presents a well-suited concept to analyze the transformative dynamics of formerly rural, peripheral or marginal places into vibrant agglomerations [1].

The relation between economic restructuring processes and urban transformation of secondary towns manifests throughout different and multiple processes [3,4]. In many contemporary African locations, economic dynamics involving both the private sector and the state, foster agglomeration in new forms of settlement in the rural areas or in rapidly urbanizing smaller towns and cities [5,6]. These processes not only involve agglomerations of people and economic activities but are also "transforming specialized land uses and the densification of networks of interactions" [7] (p. 7). Our theoretical framework is thus situated at the intersection of two academic bodies of literature that describe two key processes in the context that we analyze: the literature on mining urbanization and the literature on urbanization of forced displacement settlements. In the academic literature about the relationship between mining and processes of urbanization, the emergence of so-called 'boomtowns' in Africa has received considerable attention since the early 2000s. Mining boomtowns have historically sprung up as a result of the sudden and fast influx of large groups of fortune-seekers, attracted by the possibility of getting rich quickly. The literature, which mostly developed within the discipline of anthropology, mainly focusses on the "fast changing life worlds" in these places of "accelerated change" with increased density of connections, growing stakes and multiplying challenges and opportunities [8] (p. 43). It has been extensively documented what motives drive people to move to contemporary mining boomtowns in the Global South [9–11], as well as what the socio-economic impact of mining activities has on people living and working in or near boomtowns [12–14]. Studies have scrutinized the creative ways in which boomtown dwellers cope with the multiple uncertainties associated with the cyclical nature of commodity prices or posed by violent conflict [15–17]. Attention has also been paid to the aspects of urbanized mining (sub) cultures, which are often characterized by the creation of remarkably different lifestyles [18–20]. Some scholars have explicitly discussed the political aspects of boomtown

urbanization [21–24]. Finally, the phenomenon of "big men politics" in mining boomtowns has been discussed, among others, by Werthmann [25], Cuvelier [26] and de Koning [27].

Forced and voluntary migrations have also been in the center of the discussion about urbanization in Africa. Over the past two decades, there has been a growing academic interest in the catalyzing role of violence-induced displacement in processes of rapid urbanization. Apart from the extensive literature on the integration of refugees and Internally Displaced Persons (IDPs) in existing African cities [28–30], there has been an interesting strand of literature on the transformation of temporary refugee camps into new cities or towns. Agier [31] introduced the concept of "city-camp" or "camp-city" to study the "novel socio-spatial form between war and city" and Jansen [32] added the concept of "accidental cities" and "accidental urbanity" to analyze the spatial and socio-economic transformation of temporary "unforeseen" settlements into permanent cityscapes. Urbanity, as the term is applied in this paper, refers to the specific spatial and socio-economic conditions in these new agglomerations [22,32]. It refers to the outcome of the transformation from rural or other previously non-urban spaces such as refugee camps into realities lived and perceived as cities. Compared to the growing number of studies about urban (international) refugees [33–35], the urban situation of IDPs has received relatively limited scholarly attention until now [36,37]. A challenge in the research on urban IDPs is that most of them are "invisible" as they stay outside the camp structures and therefore are not registered [38]. The above-mentioned literature has taken us a long way in developing a better understanding of the urban outcomes of violence-induced displacement as a geographic, demographic, socio-economic and political process. It has also drawn scholarship and focus of urban research in Sub-Saharan Africa to urbanizing processes taking place in dominantly rural areas.

Based on this background literature and the issues identified that are central for the understanding of boomtowns associated to mining and conflict, the analysis of the results is centered on the processes of town booming and how they concur to a variety of individual trajectories as well as to undefined settlement. The analysis also shows how this town building is shaped by violence and forced displacement.

## 3. Contextualizing Mining and Urbanization in the Study Area

In both Angola and the DRC, mining and interlinked forced migration caused by violent conflict have characterized agglomeration and the shaping of new and old urban centers. Zambia, in turn, while not engaging in warfare (despite its history of rebellion [39], has remained in the pathway of displacement and settlement related to political processes from the neighboring countries in the last decades. Figure 1 shows the location of studied sites and the proximity of both mining and urbanization.

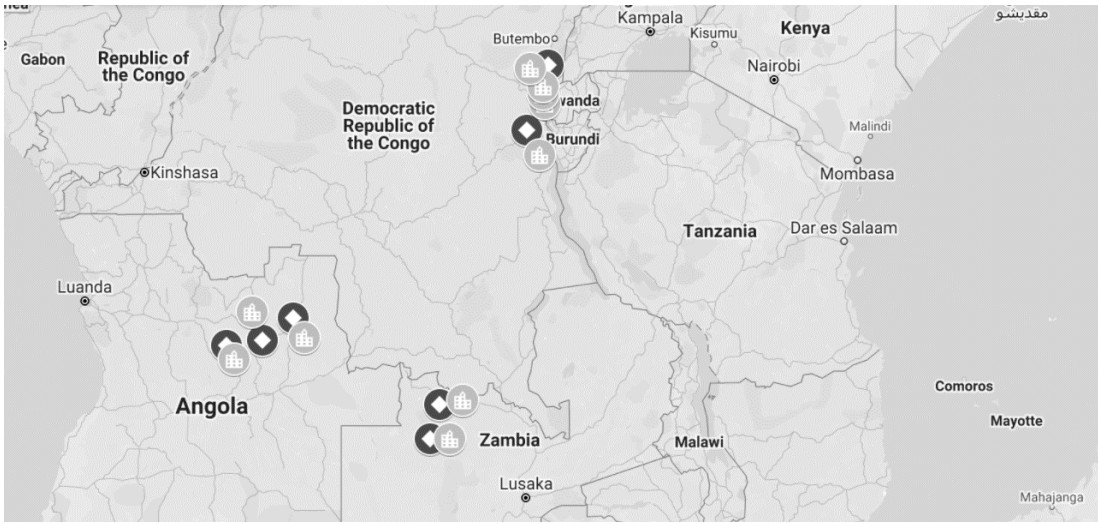

**Figure 1.** Location of resource extraction sites and urbanization. Source: Google Maps.

Mining in Zambia is associated with processes of urbanization since the establishment of industrial-scale copper mining in the 1920s. The emergence of towns in what is now known as the Zambian Copperbelt attracted the interest of scholars interested in social change and who came to influence studies of urban change and migration in Africa [40–43]. Their views of urbanization came to be later critiqued by James Ferguson [44] as presuming a unidirectional movement of development from a rural- to urban-dominated study of urban transformation in Africa. Ferguson [44–46] argued that these works missed the range and variety of ways in which Copperbelt residents associated with the countryside and the multiplicity of their identities that straddled urban and rural dispositions. Other scholars [47–49] confirming this argument indicated how livelihoods associated with the countryside came to play an important role for urban residents in a context of economic crisis, highlighting how the rural and the urban mutually constitute another, and indicate the variations in the kinds of urbanisms one encounters in the region.

In Angola, urban growth in the diamond mining regions of the Lundas (Lunda Norte and Lunda Sul provinces) is intrinsically related to major socio-political and economic transformations in the country starting in the 16th century. Colonization, until 1975, and the colonial corporate regime made mining exploration and settlement—of miners, other staff and services connected to mining—intertwined matters [50]. Diamang mining company was, for decades, in charge of both diamond mining and territory and settlement management in the Lundas. Civil war between 1975 and 2002 was characterized by the lawless presence of wartime fortune-seekers and guerrillas in the mining regions, who shaped the way settlement evolved. Boomtowns were to be seen emerging in the areas where informal artisanal *garimpo* was taking place. Peace, from 2002, and the combination of the current quasi-sovereign state of corporate mining [51], combined for many years with clandestine *garimpo*, resulting in both corporate and state-led urbanism and arrangements linked to the informal local diamond economies [52]. However, while labor migration of colonial mining was shaping the mining towns that had services and infrastructure [51], the post-independence conflict that lasted almost 30 years until 2002 was forcing migrations and immobility and creating shifty, precarious settlement.

For the Democratic Republic of the Congo, the process of mining urbanization is entangled with histories of colonial resource extraction, post-colonial political crisis, protracted armed conflict and (forced) displacement. For the region of Eastern DR Congo, where the case studies of this paper are located, mining boomtowns emerged in a context of artisanal, largely informal mining activities in settings of civil war, political instability and militarization. The Kivu provinces are extremely rich in natural resources such as gold, Colombo-tantalite (coltan) and cassiterite (tin). Almost 30 years of civil war resulted in a profound reconfiguration of local economies, livelihoods and rural-urban connections. The increased urbanization of the region can be perceived as an outcome of this transformative process [53]. Urbanization in Eastern Congo unfolded in two distinct forms: Firstly, in the increasing pressure on and expansion of established urban centers (which have their origins in the colonial period) and secondly, the mushrooming of new boomtowns that did not exist before the war [2]. With population figures between 20,000 and 100,000 inhabitants, these towns have emerged in the rural hinterlands of the Kivu provinces as a result of people's mobility in search for protection and livelihood. Some of these towns developed and 'boomed' around a refugee or internally displaced person (IDP) camp, others around mining sites and, still others, around trading axes [2]. Boomtowns in the Kivu region not only developed into dynamic centers of development, exchange and accumulation, but also into strategic nodes of armed mobilization and forced displacement [22].

Given the contexts for urbanization related to mining, we start from the fact that the three case studies under analysis demonstrate different types of outcomes influenced by different labor migration dynamics and economies, and by different conditions of forced and voluntary migrations.

## 4. Materials and Methods

Based on the authors' long-term field experience in the countries under study, the analysis draws on ethnographic fieldwork data collected through observations, focus group discussions, as well as individual and collective interviews with key actors involved in the everyday shaping of boomtown urbanism. The authors' review of the research about migration and urbanization in Angola, the DRC and Zambia is confronted with the empirical research and data to compare processes of town booming and individual and collective migration and settlement trajectories (see Table 1. for a summary).

**Table 1.** Methods and research sites.

| | Materials | Urban Sites and Population | Mining Dynamics | Conflict Dynamics |
|---|---|---|---|---|
| **Angola** | Interviews Household survey Observation | (2010 * [52]) Saurimo: 83,470 Cacolo: 15,526 Itengo: 1000 Luó: 2638 | Large-scale mining Artisanal mining | 1975–2002 Blood diamonds |
| **DRC** | Interviews, Collaborative mapping exercises Group discussions Observation | Kitchanga: 80,000 ** [2] (2017) Minembwe: 39,500 *** (2020) Rubaya: 70,000 (2017) **** [22] Numbi: 11,000 (2014 ***** [22]) Nyabibwe: 25,000 (2015 ****** [54]) | Artisanal mining | 1993–today Protracted violent conflict |
| **Zambia** | Interviews Group discussions Observation | (2010 ******* [55]) Kalengwa: 2075 Kalumbila district (formerly Solwezi West constituency): 85,505 | Artisanal mining Large-scale mining | 1976–1982 Political rebellion |

* [52]. ** [2]. Official censuses have not been carried out for decades. Numbers are estimations based on data provided by local administration, independent NGOs and local health facilities. *** Estimation provided by Burgomaster of Minembwe, September 2020. **** [22]. Official censuses have not been carried out for decades. Numbers are estimations based on data provided by local administration, independent NGOs and local health facilities. ***** [22]. Idem. ****** [55].

Cases presented on Zambia emerge from the ethnographic research of author Patience Mususa in Zambia's North Western province which was conducted variously as part of doctoral research (from 2007); as consulting researcher for a women and children's poverty study in 2008 with the ILO and UNICEF; with a USAID project in 2011–2012 on the effects of mining on program recipients; and as part of an ongoing study from 2015 on the planning dynamics of new mining towns [54]. Sites of study in North Western province Zambia include Kalengwa, a remote rural and mainly artisanal mining site and Kalumbila, a new district and mining town that emerged from the development of a large-scale mining operation by First Quantum Minerals from around 2011. Field research involved participant observation, semi-structured interviews with residents, customary, state, corporate and tertiary institutional actors, as well as focus group discussions. Semi-structured interviews on the migration stories presented in this paper were conducted with participants in Kalengwa in 2008, and in Kalumbila in 2016. The migration cases provide insight into the economic motivations of migrants, and how services and infrastructure play a role in settlement. They are demonstrative of the kinds of processes and dynamics that may lead to the consolidation of settlement in booming mining areas.

The case studies conducted in the diamond mining province of Lunda Sul in Angola by author Udelsmann Rodrigues analyzed the direct linkages between migration and mobility—and immobility—and urbanization and settlement. The empirical research took place in 2011 and was conducted within a comparative research program, Urbanisation and Poverty in Mining Africa (UPIMA) of the School of Geographical and Earth Sciences, University of Glasgow, funded by the UK's Department for International Development (DfID) and the Economic and Social Research Council (ESRC) (RES 167-25-0488). The comparative research involved three countries and main mining industries—Angola (diamonds), Ghana (gold) and Tanzania (gold and diamonds)—and aimed at analyzing key aspects of urbanization in mining regions and their relations with poverty, precarity and

instability. The research involved a household survey of over 150 households and more than 50 in-depth semi-structured interviews with a variety of local actors and stakeholders in Saurimo (capital of the province), Cacolo, Luó and Itengo. The localities vary not only in size and predominant mining-related operations—artisanal and industrial—but have also been differently exposed to the civil conflict and the exploration of 'blood diamonds'. A follow-up on the primary field research was conducted in 2012 through a series of meetings and participatory activities (photo contests with students, focus group discussions) in all four towns. In addition, the contacts established then, with enumerators and a number of key informants, were kept, namely through social media, and proved to be crucial for exchanges and discussion of research results.

The insights for boomtowns in Eastern DRC for this paper are based on qualitative research conducted by the authors Karen Büscher and Jeroen Cuvelier in collaboration with several local research collaborators since 2012. Fieldwork conducted in the town of Nyabibwe (South Kivu) focused on the distinct cultures, economies, architecture and governance of mining urbanization in a context of fierce political conflict and contestation [53]. Further comparative research on Rubaya and Numbi (data collection in 2015, 2016) investigated the urban expansion process of these towns in itself as a highly political process, involving different state and non-state actors at the local, provincial and national levels. This research demonstrated how, in a context of war, the establishment, expansion and administrative recognition of these centers represent important political, economic and social resources for multi-scalar alliances of elites, customary authorities, armed groups and the Congolese state in their broader political struggles for power, legitimacy and control [22] (p. 3). Collaborative research with Gillian Mathys on boomtown urbanization emerging from forced displacement in Kitchanga (fieldwork in 2017) has analyzed boomtown urbanization as both a 'product' as well as 'productive' of political or military struggles over power and control, and has contributed to the analytical understanding of the dynamic relationship between violent conflict and urbanization as mutually reinforcing processes. Finally, the case of Minembwe has been added to our research in 2020 with fieldwork in collaboration with GEC-SH (*Groupe de Recherche des Conflits et Sécurité Humaine*, Research Institute based in Bukavu, DRC) on the recent political contestation concerning its transformation into a *commune rurale* (cf. infra). Data has been collected through a variety of fieldwork methods, such as collaborative mapping exercises with different groups of respondents such as IDPs, traders, miners (documenting/visualizing spatial aspects of boomtown urbanization); focus group discussions with different categories of boomtown inhabitants (refugees, miners, transporters, IDPs, soldiers, investors, traders); semi-structured interviews with key administrative, political, economic and military actors with strong power positions in these boomtowns (big men, customary chiefs, administrative delegates, civil society representatives, heads of mining and commercial associations, representatives of the Congolese security forces, NGO staff); observational walks and the consultation of local administrative reports.

## 5. Results

In line with the main focus and aims of this paper, rural-urban transformation and boomtown urbanization in the three countries under study are being presented below along the following structuring lines for comparison: (i) underlying economic processes leading to the emergence and development of new urban agglomerations; (ii) consolidation trajectories defining temporary or permanent forms of urbanization and (iii) the impact of violence and forced displacement on the spatial and socio-economic characteristics of boomtown urbanization.

### 5.1. Underlying Processes of Town Booming

In Angola, war has dictated the booming of mining-related urban agglomerations after independence and reverted the dominance of large-scale mining companies for all settlement in mining regions [50,52]. While the characteristics of informal booming towns

throughout the conflict have not been particularly 'urban' given the absence of infrastructural investments or development strategy by the military that controlled these spaces [56], these unplanned settlements, which emerged in new rural areas, heavily associated with the *garimpo*, concentrated more than merely large numbers of population. The local populations, the military and cross-border migrants from the DRC [57] compounded the effervescent new agglomerations in the Lundas. The "booming diamond settlements" [56] (p. 550), where new displacement economies were flourishing [52], concentrated dynamics of urban living, consumption, and, above all, of intense commercial and trading activities. After the end of the civil war, these boomtowns "resumed their status of government and mining company-dominated settlements, where control and planning prevail" [52] (p. 687). The new mining law of 2011 that aimed at regulating informal *garimpo* and settlement related to it clearly targeted returning the control of the diamond economies to the government but also of the emerging towns. Angola's urban growth over the last decades has been characterized by rapid agglomerations in existing towns and cities—and in the capital Luanda in particular—as a result of the civil war playing out in the countryside. The post-war tendency, however, has been of state-led urbanism, with an emphasis on major infrastructural investments [58–60]. The two main types of urban emergence and growth—the expansion of existing towns and the *garimpo* boomtowns—combine dynamics of both formal and informal construction and a variety of scales of urban dynamisms.

A surge in mining investment since the early 2000s as a result of growing demand for copper/cobalt has seen the emergence of large-scale and small-scale mining in Africa's Copperbelt of Zambia and the Democratic Republic of Congo. This is occurring in predominantly rural areas, such as in Zambia's North Western province [61]. These dynamics are changing the character of places where residents have mainly been making a living from small-scale agriculture, hunting and gathering, and some trade in agriculture and forest produce with adjacent urban and border regions [62,63]. Mining has attracted new migrants. Many are coming from the older, established mining towns of the Zambian Copperbelt, where the withdrawal of state welfare that came with deregulation of the economy and the reprivatization of the state conglomerate, the Zambia Consolidated Copper Mines (ZCCM) from the mid-1990s, saw massive job layoffs and a protracted economic crisis [49]. A dual process of mining urbanization is occurring alongside both large-scale and artisanal mining. It is catalyzing demand for goods, services and housing both for short- and longer-term stays. At artisanal and small-scale mining operations, and on the fringes of the planned towns for large-scale mining, settlements that do not lend to easy description of rural or urban have emerged. Their ambiguous character can be described as 'rurban', connoting the interconnections and social political transitions between countryside and city [5].

The North and South Kivu provinces in Eastern Congo are not only characterized by the abundance of natural resources, but also by the mobilization of armed groups, strong ethnic tensions, violent struggles over public authority and militarized land conflicts. In the early years of the new millennium, during the RCD rebellion that coincided with the global boom in demand for coltan and cassiterite, several boomtowns emerged around artisanal mining sites in these provinces. Over the last two decades, towns like Numbi, Nyabibwe or Rubaya have experienced spectacular growth, transforming from small villages into extended 'urbanized' agglomerations. The substantial influx of refugees and IDPs further pushed their demographic expansion in the context of war. The thriving forces behind the fast expansion of these mining agglomerations are multiple, and a combination of structural processes and people's agency. The more these boomtowns expand, the more miners and IDPs have been followed by an influx of all kinds of other 'migrants' seeking to integrate into and invest in these emerging semi-urban economies. This has led (in some cases more than others) to the creation of an ethnically and economically diverse demography, the development of new markets, increased circulation of money and goods and the spontaneous development of infrastructure and services. The urbanization process may initially unfold in an informal way, yet various actors (such as economic big men or political elites)

have increasingly tried to incorporate this process into their own political or economic ambitions [22]. While in some cases, local influential actors explicitly invest in these towns, in other cases, rumors circulate about elites boycotting investments in urban expansion to prevent external actors from interference. The transformation of peripheral rural sites into urbanized centers of (largely informal) accumulation and survival has resulted in a profound socio-economic, spatial, but also political reorganization. Urbanization comes with increasing presence of state services, which often results in a 'double' governance system, whereby customary (traditional) authority and state (administrative) authority are exercised simultaneously, often in competition. In light of regional (violent) ethnic politics, this fragmented leadership has been often conflictual.

### 5.2. Town Booming Trajectories and Undefined Settlement

In Angola, accounts of forced migration and settlement in mining regions have been concentrated in the provinces of the Lundas. Here, the conditions dictated by the developments of the civil war made the life of the residents constantly uncertain, straddling between the rural and the urban. The civil war created the conditions for the persistence of undefined makeshift settlement for many years in the booming mining agglomerations while the 'safer' cities in the country and abroad were the alternative in particularly 'unbearable' periods of the civil war. Accounts collected in the Lunda Sul frequently refer to the unclear and unpredictable developments of the war linked to undefined conditions for settlement. The realities lived there during these periods were also quickly changeable and unclear. The thriving economies developed during the war involved intense circulations and a variety of businesses. M.S. mentioned that there were traders from Zaire based here that directly exchanged products for diamonds with the population; no money was even used. These traders brought clothes, guns, and other commodities (M.S., Cacolo, July 2011). The existence of the war and of what it would dictate in terms of settlement and local socio-economic dynamics generated unclear and uncertain settlement and also led to displacements of varied extension. As S. accounted, he had come back to Cacolo in 1994 when his father (since colonial times, a businessman in the region, in Cacolo) needed him to come back from Brazil where he had lived "because one could not cope with the war. I left Cacolo in 1998 inside an army tank" (S., Luanda, August 2011). Unsteady settlement and living were also, at specific points in time, the reality of varying numbers of people in the Lunda region. For instance, M. referred that the mining town of Lucapa, that had been projected for 10,000 people, had, between 1992 and 1994, 100,000 inhabitants living there because the government "closed the eyes" to the *garimpo* (M., Luanda, 18 August 2011). The region's boomtowns are then characterized by the expansion and contraction of agglomerations, the intense movements between the rural and the urban and by the unpredictability of urban consolidation.

Kalengwa in Zambia, is a mining outpost whose settlement dynamics have changed in relation to the country's copper fortunes. The state mining conglomerate ZCCM had run the satellite mine there until 1982 when low copper prices had made it financially unfeasible to continue operations. Kalengwa had been managed as a mining camp in contrast to the company towns on the Copperbelt province that the mines administered. When mining operations ceased, the population in the area declined due to the breakdown in basic services such as water, and the absence of wider employment opportunities. In the aftermath of the reprivatization of ZCCM, Kalengwa mine was sold and embroiled in an ownership dispute. Following the 2000s' commodity boom, Kalengwa saw, from around 2007, a growth in the local population as newcomers arrived in the area to engage in artisanal mining and other economic activities [64]. Some came from other rural areas in the region such as Kabompo district; others from the urban centers of Copperbelt and the capital city, Lusaka. Newcomers comprised men, women and children engaged in artisanal mining, trade, brewing beer, and the provision of services, such as the collection of water. In line with our ethnographic approach, to illustrate migration and settlement trajectories into the town, we briefly introduce some individual narratives: S.B., a woman in her thirties



(interviewed on 4 August 2008) had moved to the area with her husband and four children from an informal settlement in Lusaka. S.B.'s husband, J.B., had been the first to move in 2007 to informally work at the mine. From the proceeds of mining, he had bought pigs to resell in Lusaka. After an initial period of moving back and forth between Kalengwa and Lusaka to visit his family, his wife S.B. and the children joined him in Kalengwa. S.B. saw an opportunity in brewing beer for sale, and did quite well with the business. After failing to get the children into the local public school which became crowded out, her children stayed with her at the makeshift structure that made for a temporary home, where they helped to carry and sort copper ore. S.B. and her husband were building an earth block house in the village, and scouting for agricultural land in the adjacent, more fertile areas, to consolidate their ties in the area, but they also maintained their base in Lusaka as it was good for trade.

The straddling of life between the mining outpost and established urban areas was not unique to artisanal mining sites. It also applied to places where large-scale mining was happening, like in Kalumbila (formerly Solwezi West) where, despite well-developed infrastructure and the availability of a growing number of formal jobs, mainly in mining, residents still maintained ties with older established urban areas. An illustration of this is B.C. (interviewed in Kalumbila on 26 July 2016), born in 1990, and trained as a mechanical engineer in Kitwe, a town on the Copperbelt, and was a job-seeker living with his older brother L.C. They lived in a two-bedroom architect-designed house with a landscaped garden, which they shared with two other subletting tenants. One was a male electrician, B.C.'s age, and was working for a construction company; and the other a female, aged 29 years, working as a shopkeeper. The house was a rental, with an option to buy, an initiative the company was promoting to encourage long-term settlement. L.C. had been working in Kalumbila as a mechanical fitter for the mines since 2013 and was, at the time, engaged to be married, though his fiancée had put off moving to the town until she completed her teaching training on the Copperbelt. To earn a living while looking for a job, B.C., despite the irregular transport links between Kalumbila and the nearest big town, Solwezi, travelled monthly to the Copperbelt to buy groceries for resale, but also to keep an eye on other job opportunities; he was also saving up money to buy a stove so he could bake scones to sell. In line with the multiple livelihood options that residents from the Copperbelt had taken up following the economic crisis, his brother L.C. had acquired a piece of land in the nearby rural area and planned to establish a farm. Kalumbila's mine township population at the time was largely male as formal jobs for women in mining and construction were limited. It meant that some residents were reluctant for their wives to move to the town, especially as some already had established businesses on the Copperbelt. Those with children were also concerned about the quality of education in the region so kept their children in schools on the Copperbelt and maintained dual households. Thus, despite the visible urban infrastructure that was being developed in the mining town, the lag in services, and lack of diversity in economic opportunities effectively made for less stabilized residents.

The Zambian situation contrasts with the specific context of instability and violence in Eastern DRC that strongly increases people's mobility and the 'undefined' status of people's settlements. Against a backdrop of protracted and repeated displacement, IDPs who install in these boomtowns largely live in limbo. In the case of mining boomtowns, the fluctuating prices and governance regulations of natural resources may also increase people's mobility. The 'accidental' urbanity produced in this context strikes a balance between safety and contingency, and is shaped by the tension between temporality and permanency. It is clear that many IDPs perceive their stay as transitory. However, in a situation where after almost three decades of war displacement has become a permanent state of being for thousands of people, any sense of temporariness is nuanced [65]. Long-term displacement may lead to long-term engagement, creating permanent forms of urbanization. Urbanized environments, with their extended educational facilities, healthcare, infrastructure, transport, markets, and culture, can provide an appealing environment to

broaden one's horizons. In some boomtowns (like Kitchanga in North Kivu for instance), the permanent settlement of IDPs has been documented to be politically influenced. In the context of the fierce struggle over land and public authority along ethnic lines, the 'politics of presence' and the ambition for ethnic majority has led, for example, armed groups and local authorities to distribute land to IDPs and to push them to permanently settle in these towns [2]. Additionally, the process of town booming is further politicized by the current Congolese decentralization reform, whereby several boomtowns are in the process of changing their administrative status from 'village' into '*commune rurale*'. According to the decentralization reform, agglomerations change administrative status according to their demographic statistics, and since 2013, a number of boomtowns in different provinces have been listed for such an 'upgrade' under condition that they meet a number of criteria (like minimum urban infrastructure). In the current context of violence, ethnic tensions and a confusing situation of a half-implemented decentralization law, this administrative 'upgrade' of boomtowns becomes a highly political and heavily contested process. Local elites in these boomtowns mobilize along provincial and national political levels to either push for or boycott the official recognition of their agglomeration as a *commune rurale*. This recognition implies a number of important political/administrative shifts, by which state authorities in power will be locally elected instead of appointed (and thus inevitably represent the ethnic majority of the town), and by which customary authorities will become largely irrelevant and lose their public authority (and capacity to levy taxes, for instance). In some cases, like Minembwe in South Kivu, the installation of the *commune rurale* led to a national crisis, when the installation of the new burgomaster was perceived by parts of the local population as a 'coup' by one ethnic group to dominate over others. Fueled by historical narratives of 'balkanization', and further reinforced by ongoing dynamics of violent struggles over land and political representation, the administrative recognition of Minembwe resulted in an escalation of violence, and an eventual intervention by the President of the Republic himself who cancelled the process (https://www.bbc.com/afrique/region-54515096, accessed on 12 February 2021). In other cases, like the mining town of Rubaya in North Kivu, it has been observed how in the light of this decentralization process, local elites intensively invest in their agglomeration for the town to be legible to the status of *commune rurale*. Simultaneously, they mobilize their political connections at the provincial and national level to secure a key political position in the future governance structure of the town. The act of 'making permanent' temporary urban settlements like mining towns or former IDP camps is, as such, an explicit political engagement, as are the attempts to (sometimes violently) boycott the process to prevent this permanent status.

*5.3. Urbanity Shaped by Dynamics of Violence and Forced Displacement*

As emphasized, Angola's civil war dictated the flows, directions and durations of migrations and settlement. This was particularly key in defining and shaping the dynamics of urban emergence and of town building in the Lundas. The absence of a national census between 1970 and 2014 makes it difficult to trace population estimates and indirect accounts point to intensified rural-urban movement, particularly in the studied towns of Saurimo, Cacolo, Luó and Itengo [24]. Massive migration to the 'safe' cities controlled by the government during the war took place at the same time as artisanal mining booming settlements were appearing under the control of the UNITA guerrilla and the influence of external markets, migrations and a series of 'blood' diamond mining-related businesses. These dynamics made population fluctuate in size, attracting then repelling in-migration, and creating different conditions for migration, at times forced and in other occasions, voluntary, like in the case of diamond 'luck' seeking. M.S. was born in 1938 in Xassengue and became a teacher in Cacolo in 1961 and a deputy Municipal Administrator of Cacolo in 1986. He mentioned that after independence, "many people from Xassengue, Cucumbi and Alto Chicapa came to Cacolo because of the war, especially in 1983." Cacolo, a town built in colonial times, has expanded and contracted the number of residents agglomerating

there as the war evolved. During the war, the government and the UNITA guerrilla would alternatively occupy the town for two to three months and during these occupations, the population that could, would further move to the capital of the province, Saurimo. He mentions that "in August 1995, we walked to Saurimo to escape the war. Thirteen died on the way. Some remained in Cacolo, in captivity (*cativeiro*) because they could not reach Saurimo."

While Zambia does not have much of a recent history of conflict-related dispossession—except as a receiving country of refugees from neighboring Angola and the DRC [66]—forced relocations, in particular of rural residents, have been connected to large-scale developments, such as in the seminal work of Elizabeth Colson [67] on the building of the Kariba Dam in the 1950s to industrial mining at the turn of the 20th century on the Copperbelt, and more recently, since the 2000s, in North Western province to emergence of new mines. As Lisa Cligett and others [68], following up decades later on the experiences of residents who were forced to relocate with the building of the Kariba dam, socio-economic effects across generations on those relocated can create a protracted situation of chronic precarity. Residents of the chieftaincy of Musele who paved way for the mining development in Kalumbila were well aware of the long-term consequences of the mine on their livelihoods and ways of life. For those who had to move and who lost their gardens and were compensated under a combination of both national and international mining guidelines, for them, the compensation packages could not replace the sustained livelihood that lifelong access to land could. This led, for some, in the planned resettlement area, and others within the densifying villages to uproot to the further distant rural regions in the province.

The specific context of the war-torn DRC results in the interesting, contradicting image of boomtowns as both 'safe havens' as well as 'rebel hotspots'. In Congo's Kivu provinces, urbanized sites embody 'safe' spatial environments for IDPs or refugees, at the same time, being safe havens or strongholds or strategic targets of armed groups [65]. As a safe haven, boomtowns attract refugees and IDPs because of the relative presence of security forces, infrastructure and international actors such as MONUSCO and NGOs. The presence of IDPs has become a common characteristic of boomtowns in the Kivu provinces. In mining towns like Nyabibwe, Rubaya or Numbi, IDPs settled within the mining agglomeration where some of them went to work in the mines, others integrating into other livelihoods. Other boomtowns, like Kitchanga, evolved from the gradual urbanization of IDP camps, where forced displacement was the main driver of the town's expansion. The impact of the proliferation of armed groups in the Kivu provinces on boomtown urbanization has been described in earlier research [22]. Boomtown expansion in a context of war in the Kivu provinces has generated an urbanization process, which is profoundly militarized. This militarization translates itself not only by the physical presence of armed groups or the formal Congolese security forces, but also by the vast proliferation of arms among its inhabitants, the relatively easy access to violence to settle disputes or the public authority of military actors such as generals or colonels investing in the town's economy and real estate. These tendencies make boomtown urbanization a process that is often conflictual. As the example of Minembwe in this paper or Kitchanga in earlier research demonstrates: boomtown urbanization can trigger conflict and violence. Without being the main cause of violence, the process of boomtown expansion can reinforce historic struggles over land, citizenship and public authority along ethnic lines.

## 6. Discussion

As centers of attraction, opportunities, aspiring urbanities and diversified livelihoods and cultures, boomtowns represent fascinating sites from which to investigate rural and urban transformation in a context of resource extraction and conflict/post conflict governance. The findings point to discernible patterns of boomtown consolidation across these adjacent countries, which are a result of combinations of types of migration, migrants'

agency and the local governance structures. The main feature of town booming across the case studies is the fact that this 'undefined' settlement tends to prevail for a long time.

In Angola, indirect urbanization in already established cities and towns, like in other urban mining contexts in Africa [69], followed the trend for urban 'informal' expansion to rural areas, which is characterized by precarity and temporariness. Boomtowns in the countryside, stimulated by the diamond explorations during the civil war, have also followed a pattern of precarity and were never perceived by the migrants and fomenters of agglomeration as long-term projects. Angola's post-war growing state-led urbanism envisages reverting the indefinitions that have characterized settlement in the last decades, aiming at the consolidation of agglomerations into 'proper' cities, as stated in the country's urban plans, and visible in the post-war reconstruction of cities and towns. The DRC is particularly characterized by informal urbanization, yet highly politicized. This urbanization process is inevitably interwoven with agendas of political, military and ethnic mobilization. To recognize boomtown consolidation as not only a spatial, socio-economic and demographic but also a political process, it is important to understand the boomtown as the spatial arena of political constellations and to understand the process of urbanization as a part of political mechanisms. Zambia presents a case where urban consolidation and migration has been strongly related to the emergence of large-scale mining industry. Contemporary processes of migration and settlement are still dominated by the effects of large-scale mining, whether in boom or bust, but as with trends elsewhere, new processes of settlement connected to artisanal mining in rural areas are also creating new kinds of urbanisms that maintain aspects of the rural and urban. What is also emerging, and requires further study, are the varied ways in which migrants in these new mining areas are, for mainly economic opportunities, straddling a living across some well-established urban areas, but also other rural regions. These somewhat itinerant settlement processes, as shown in the cases on Kalengwa and Kalumbila, indicate a mobile pattern of migration that nevertheless seeks to consolidate itself in place, even if it be in multiple places. The constraints of the time-limited country-focused researches used for this comparative analysis pose challenges to the understanding of processes that constantly reconfigure urban and rural spaces over time and, at the same time, tend to lose the cross-border regional dynamics of circulation and settlement. Better comprehensive approaches and research are therefore desired, particularly in contexts of precarity and instability, such as the region under analysis.

## 7. Conclusions

This study is the result of combined insights from Zambia, Angola and the DRC into the phenomenon of rural-urban transformation and boomtown urbanization. More particularly, our article has presented an in-depth qualitative analysis of the impact of the combined economic and political processes of mining and violent conflict on the characteristics of urbanity emerging in these new settlements.

Boomtowns in rural mining Africa create fast-changing types of living locally and generate multiple and varied responses from the urban dwellers, often mobilizing informal responses, depending on the systems of governance and the developments in the extractive industries. The case studies have shown that these processes are catalyzed by violent, induced displacements and emplacements and by radical transformations of macro-scale economies or political reforms. Agglomerations in rural boomtowns remain for longer or shorter periods of time, transitory, undefined proto-urban spaces, and consolidation, becoming city, is dependent on a combination of factors, among which the structures of governance and residents' agency appear as key.

The availability and quality of the physical infrastructure of these booming places may lend to the sense of the provisional seen in makeshift structures. However, it is not only this that lends to the tentative quality of some of the processes of urbanization being observed—it is also the situating of economic opportunity in both rural and urban modes of livelihood. In addition, the complex and sometimes competing governance structures

of various actors may sometimes inhibit a visible consolidation of settlement. This leads to the need for a more nuanced understanding of the urban dynamics shaping African settlements. New urban agglomerations emerging in fast changing economic and political regions represent, at the same time, central nodes of opportunity as well as contestation. They merit increasing attention by academic as well as governance actors in order to fully understand and recognize their future potential in processes of development and stability.

**Author Contributions:** Conceptualization, C.U.R., P.M. and K.B.; formal analysis, C.U.R., P.M., K.B. and J.C.; funding acquisition, C.U.R.; investigation, C.U.R., P.M. and K.B.; methodology, C.U.R., P.M. and K.B.; project administration, C.U.R.; supervision, C.U.R., P.M. and K.B.; writing—original draft, C.U.R., P.M., K.B. and J.C.; writing—review and editing, C.U.R., P.M. and K.B. All authors have read and agreed to the published version of the manuscript.

**Funding:** This research was funded by the Riksbankens Jubileumsfond, grant number P19-0271:1 and FWO grant number G0B7821N.

**Institutional Review Board Statement:** Not applicable.

**Informed Consent Statement:** Not applicable.

**Data Availability Statement:** Data sharing is not applicable to this article.

**Conflicts of Interest:** The authors declare no conflict of interest.

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
