# Peer review of "Boomtown Urbanization and Rural-Urban Transformation in Mining and Conflict Regions in Angola, the DRC and Zambia"

_sustainability, doi:10.3390/su13042285_

Round 1

Reviewer 1 Report

The subject of the publication is interesting. However, the peer-reviewed article does not have the characteristics of a scientific publication. Research methods were not indicated. The rules of the conducted research were not defined. Only places have been identified. Individual cases are simply described as in popular science publications.

Author Response

We would like to thank the individual readers and the editors for their time and effort in reviewing our draft article in detail and for providing comments and suggestions which have been very useful in strongly improving our article.

The authors have further developed the description of the methodology section, namely on page 5, where a summary table was inserted.

Reviewer 2 Report

The paper demonstrates in a very interesting way the process of transformation of areas with different degrees of urbanization in Central and Southern Africa. Well-chosen structure of the paper, purposes, methods and comparative case studies perfectly illustrate this process. The research is well described. The conclusions of the research have a cognitive value useful for further study. One suggestion: the paper should be supplemented with graphic illustrations showing the location of described case studies, in a context of region, resource extraction or refugee camps. The authors might also consider including maps showing the urbanization transformations that have occurred in selected areas over the years. This is just a suggestion to consider, but it is not necessary for the final publication.

Author Response

We would like to thank the individual readers and the editors for their time and effort in reviewing our draft article in detail and for providing comments and suggestions which have been very useful in strongly improving our article.

The authors have inserted/annexed a map with locations of mining and urbanization in the three countries.

Reviewer 3 Report

Dear authors, hope that my suggestion to be helpful in this minor-moderate revision of this paper. 

Author Response

We would like to thank the individual readers and the editors for their time and effort in reviewing our draft article in detail and for providing comments and suggestions which have been very useful in strongly improving our article.

The authors have further developed the Methods section, namely with a table on page 5.

Responses to the additional aspects brought in the reviewer #3 letter:

  • The theoretical section has been separated from the introduction.
  • The introduction has been re-written indicating better the aims, objectives, and main relevance.
  • The concept of boomtowns has been elaborated better in the theoretical section
  • The study area has been better and more explicitly defined, a map has been added.
  • The suggested reference has been integrated into the literature review section

Reviewer 4 Report

The authors examine transformations that have been taking place throughout the process of so-called ‘boomtown’ urbanization in Central and Southern Africa. It was based on data collected in Angola, Zambia and the Democratic Republic of Congo, adjacent countries with historical, political, regional and socioeconomic particularities.

The strength of the article is the topic and the level of language. The weakness are connected with the introduction, the structure and the presentation of the section Introduction, Materials and Methods and Results.

Please find the specific comments below.

I suggest divide the introduction with the part where author introduce to the topic – with the goal of the paper and the characteristic of the research area (the good idea is to prove that the process appear in selected countries and to part which is devoted to literature review.

I suggest lines to 47 locate to introduction and formulate aim and hypothesis. I suggest adding in this part explanation why countries were chosen. I suggest adding in this part short information about applied methods.  

I suggest in part: Materials and Methods present a table where the methods and techniques of research in particular countries were conducted. Methods should be better described if the article should be published in scientific journal. When reading this version, I got lost. If research were conducted by different authors the methods/techniques allow to compare them?

I suggest characterizing the chosen cities somewhere – when are located, number of residents, etc.  Please remember that readers may be not familiar with these areas. I understand that the authors present that information in results section, but basic information are needed.

In section: result there are missing the much more proper way of the prestation the result of the research which are described in the section Materials and Methods. I suggest presenting table where the comparison of the process which in selected countries took place. Then I suggest description and the comparison. I do not understand why information about only few residents are presented.

I suggest name the section discussion: conclusion and discussion – there much more conclusion that discussion. There are lack of information about the limitations of the research/the article indicated by authors.

Author Response

We would like to thank the individual readers and the editors for their time and effort in reviewing our draft article in detail and for providing comments and suggestions which have been very useful in strongly improving our article.

  • The sections Introduction, Methods, and Results have been restructured for clarity.
  • The Introduction was divided, with the literature section now separated (page 2).
  • A short mention to methods was introduced in the Introduction as well as justification why case-studies were chosen (page 2).
  • A table was prepared with information about the methods, research sites, and their characteristics (page 5) for clarity.
  • The section ‘results’ has been introduced better, referring to the main research questions introduced in the introduction.
  • Given the ethnographic approach of the paper, individual narratives are kept but we argued for this more explicitly in the text.
  • Sections Discussion and Conclusion were detached for clarity. Limitations of the research were also mentioned in the Discussion section.

Round 2

Reviewer 1 Report

I accept the manuscript in this form.

Author Response

NA

Reviewer 4 Report

The article after revision is much better and it does not seem to need any changes. Congratulation

Author Response

NA